# A Comprehensive Descriptive Epidemiological and Clinical Analysis of SARS-CoV-2 in West-Mexico during COVID-19 Pandemic 2020

**DOI:** 10.3390/ijerph182010644

**Published:** 2021-10-11

**Authors:** Oliver Viera-Segura, Natali Vega-Magaña, Mariel García-Chagollán, Marcela Peña-Rodríguez, Germán Muñoz-Sánchez, Ahtziri Socorro Carranza-Aranda, Iris Monserrat Llamas-Covarrubias, Moisés Ramos-Solano, Jesús Mora-Mora, Carlos Daniel Díaz-Palomera, Gabriela Espinoza-De León, José Sergio Zepeda-Nuño, Enrique Santillán-López, Samuel García-Arellano, Christian David Hernández-Silva, Darbi Alfredo Zerpa-Hernandez, Guillermina Muñoz-Rios, J. Samael Rodríguez-Sanabria, José Francisco Muñoz-Valle

**Affiliations:** 1Laboratorio de Diagnóstico de Enfermedades Emergentes y Reemergentes, Departamento de Microbiología y Patología, Centro Universitario de Ciencias de la Salud, Universidad de Guadalajara, Guadalajara 44340, Mexico; o.vierasegura@gmail.com (O.V.-S.); alejandra.vega@academicos.udg.mx (N.V.-M.); marcee24.p.r@gmail.com (M.P.-R.); jesus.moram@academicos.udg.mx (J.M.-M.); daniel.qfb.farma@gmail.com (C.D.D.-P.); nohemi_90@hotmail.com (G.E.-D.L.); guillermina@academicos.udg.mx (G.M.-R.); 2Instituto de Investigación de Ciencias Biomédicas, Centro Universitario de Ciencias de la Salud, Universidad de Guadalajara, Guadalajara 44340, Mexico; chagollan@academicos.udg.mx (M.G.-C.); samuel.garcia4566@academicos.udg.mx (S.G.-A.); dazerh69@gmail.com (D.A.Z.-H.); 3Centro de Investigación y Diagnóstico de Patologia, Departamento de Microbiología y Patología, Centro Universitario de Ciencias de la Salud, Universidad de Guadalajara, Guadalajara 44340, Mexico; jsergio.zepeda@academicos.udg.mx; 4Doctorado en Ciencias Biomédicas, Centro Universitario de Ciencias de la Salud, Universidad de Guadalajara, Guadalajara 44340, Mexico; german.munoz2599@alumnos.udg.mx (G.M.-S.); ahtziricarranza19@gmail.com (A.S.C.-A.); enrique_salo@hotmail.com (E.S.-L.); hernandezsilva7@gmail.com (C.D.H.-S.); 5Departamento de Biología Molecular y Genómica, Centro Universitario de Ciencias de la Salud, Universidad de Guadalajara, Guadalajara 44340, Jalisco, CP, Mexico; irism.llamas@gmail.com; 6INICIA, CUCS Universidad de Guadalajara, Guadalajara 44340, Mexico; biolog.moises@gmail.com; 7Maestría en Ciencias Médicas, Universidad de Colima, Colima 28040, Mexico; 8Doctorado en Ciencias en Biología Molecular en Medicina, Centro Universitario de Ciencias de la Salud, Universidad de Guadalajara, Guadalajara 44340, Mexico; samael.rodsan@gmail.com; 9Doctorado en Ciencias Médicas, Universidad de Colima, Colima 280440, Mexico

**Keywords:** SARS-CoV-2, COVID-19, epidemiology, risk factors, comorbidities

## Abstract

This study aimed to summarize the epidemiological and clinical characteristics of COVID-19 from Western Mexico people during 2020. A retrospective analysis from an electronic database of people visiting a sentinel center for molecular SARS-CoV-2 confirmatory diagnosis by RT-PCR from April to December 2020 was carried out for epidemiological and clinical description of COVID-19. Out of 23,211 patients evaluated, 6918 (29.8%) were confirmed for SARS-CoV-2 infection (mean age 38.5 ± 13.99), mostly females (53.8%). Comorbidities, such as diabetes (34.7%), obesity (31.15%), and hypertension (31.8%), presented an increased odds OR = 1.27, CI = 1.14–1.41; OR = 1.08, CI = 1.01–1.16; and OR = 1.09, CI = 0.99–1.19, respectively, for viral-infection. Moreover, fever, headache, and dry cough were the most frequent symptoms. No infection difference among sex was found. Those patients >60 years old were prone to COVID-19 severity (OR = 3.59, CI = 2.10–6.14), evaluated by the number of manifested symptoms, increasing with age. In conclusion, a high SARS-CoV-2 prevalence was found in Western Mexico. Comorbidities were frequent in infected people; nevertheless, no association with disease outcomes was observed, in contrast with the highest disease severity risk found in older patients; however, continuous monitoring should be carried since comorbidities have been reported as aggravating factors. This study can help the health officials for the elaboration of planning efforts of the disease management and others in the future.

## 1. Introduction

At the end of 2019, emerged in Wuhan, Hubei Province, China, an outbreak of severe respiratory illness of unknown etiology [1]. Later, it was reported by the WHO that this novel type of coronavirus is a novel coronavirus-2019, named severe acute respiratory syndrome coronavirus (SARS-CoV-2), and is responsible for the outbreak [2]. Phylogenetic analysis indicates that SARS-CoV-2 genome shares 79.5% and 50% sequence identity to SARS-CoV and MERS-CoV, respectively [2]. COVID-19, the name of the disease caused by SARS-CoV-2, shows a mild course in 80% of observed patients and a severe course in 20%, with a worldwide lethality rate of 2.05%, according to Johns Hopkins coronavirus resource center [3]. The first confirmed case of COVID-19 in Mexico was on 27 February 2020 [4], and until 14 March, the first two cases in Jalisco state, the fourth state with the major population in the country [5]. As of September, there were 3,645,599 confirmed cases and 276,376 deaths in México, while 153,237 cases and 16,070 deaths were reported in Jalisco state, positioning Mexico among the five countries with the most deaths proportionally to their COVID-19 cases, with a reported cumulative case fatality rate of 7.6% [3].

The reported symptoms associated with COVID-19 are headache, anosmia, fever, cough, nasal congestion, and fatigue, among others [6]. COVID-19 has been reported across all age groups; elderly people and people with underlying diseases have a higher risk of getting severely ill and need intensive care [7]. Some underlying diseases reported to affect the most are obesity, diabetes mellitus, cardiovascular disease, and chronic kidney disease; furthermore, the increase in mortality is mainly associated with cardiovascular diseases. Therefore, the prevalence of obesity (36.1%), diabetes (10.3%) [8], and cardiovascular disease (22.8%) [9] in Mexico are crucial factors in the development and management of this pandemic in the country. COVID-19 has had a different impact worldwide, affecting some countries in a more severe course than others. The epidemiology differs from country to country, depending not only on the disease, but also on differences in case detection, testing, age range, economic activity/sector, underlying diseases, and implementation of public health measures [10].

Several efforts have been employed to understand the ongoing situation and the development of strategies to contain the virus. The first contention measures in Jalisco state were implemented on 21 March 2020 [11]. In fact, diverse diagnostic kits to evaluate COVID-19 are available, and several repurposing therapeutics have been shown to be clinically effective for COVID-19 [10]; however, the SARS-CoV-2 infection is still spreading, causing serious concerns to public health and the economy.

Despite the availability and application of several worldwide vaccine alternatives, COVID-19 continues to represent a threat to the global health system. It was important in this study to examine and analyze the epidemiology of the coronavirus disease (COVID-19) through a sentinel center open for the general public focused on the SARS-Cov-2 diagnosis in Jalisco, Mexico, during the first months of the pandemic in the state from April to December of 2020.

## 2. Materials and Methods

### 2.1. Study Population

Data herein were retrieved retrospectively by analyzing an electronic database constructed transversely from people assisting the Laboratorio de Diagnóstico de Enfermedades Emergentes y Reemergentes (LaDEER) for confirmatory real-time PCR (RT-PCR) SARS-CoV-2 diagnosis during the April–December 2020 COVID-19 pandemic. LaDEER, located in the Universidad de Guadalajara, operates as an ambulatory diagnosis sentinel center for COVID-19 monitoring in Jalisco state, Mexico, from April to current. According to the epidemiological standards, a molecular diagnosis was performed from an oro-and nasopharynx swab in individuals manifesting symptoms related to COVID-19 and those in contact with patients previously diagnosed with SARS-CoV-2 in the last fifteen days.

All the clinical and epidemiological information from people with suspected SARS-CoV-2 infection was entirely recovered in a database containing personal information, patient residential address, comorbidities, symptoms, travel history, and economic activity. The economic activity was classified into sectors according to the Instituto Nacional de Estadística y Geografía [12], as primary as the exploitation of natural resources like forestry, agriculture, mining, and quarrying; secondary as manufacturing, processing, and construction (infrastructure) industries; and tertiary as goods distribution, information transactions, transactions with assets, services whose primary input is the knowledge and experience of personnel, recreation-related services, and other [12]. In addition, a telephonic survey, designed and conducted in collaboration with health institutions for epidemiological surveillance, obtained the data prior to sample collection for RT-PCR analysis.

### 2.2. Epidemiological Surveillance by RT-PCR for SARS-CoV-2 Detection

During April–December 2020, all patients with suspected SARS-CoV-2 infection were screened by RT-PCR using either the DeCoV19 Kit Triplex or WoV19 Kit (Genes2life S.A.P.I. de C.V., Irapuato city, Mexico). Both kits are based in the CDC as diagnostic panels for SARS-CoV-2 detection and received in the early stages of the pandemic the emergency use authorization from the Instituto de Diagnóstico y Referencia Epidemiológicos (InDRE, Ciudad de México, Mexico) and the Comisión Federal para la Protección contra Riesgos Sanitarios (COFEPRIS, Ciudad de México, Mexico). DeCoV19 Kit Triplex targets three nucleocapsids (N) gene regions; meanwhile, WoV19 Kit identifies a region inner gene E and RdRp. The kits also include a primer mix to detect the human RNase P gene. A sample was considered positive when an exponential growth curve with a Ct-value was below 35. All analyses were accomplished in a BSL-2 laboratory.

### 2.3. Statistical Analysis

The categorical (qualitative) variables were summarized as frequencies and percentages, while continuous (quantitative) variables as mean ± standard deviation and median and percentiles according to the data distribution evaluated by a Kolmogorov–Smirnov test. Comparative analyses for groups were assessed with Student’s *t*-test and variance analysis for variables normally distributed. For nonparametric data, a Mann–Whitney U was implemented. A Chi-square and McNemar test was used to compare proportions among groups. Furthermore, a univariate test was achieved for groups with a significant result as appropriate. A *p*-value of <0.05 was considered statistically significant.

## 3. Results

### 3.1. Demographic Information, Underlying Diseases and Travel History of the Overall Population

Between April and December 2020, we screened 23,211 suspected cases of SARS-CoV-2 infection from swab specimens from subjects in Western Mexico. The mean age was 37.4 ± 14.2 years, from which 53.6% were female and 46.4% were male. As shown in Table 1, 6918 (29.8%) were positive for SARS-CoV-2 infection, presenting a mean age of 38.4 ± 13.9; most (53.6%) were female. (*p* < 0.001). The higher proportion distribution of people assisting the diagnosis center was between 20 and 59 years old (Figure 1). Patients with age ranging from 30–39 showed the largest absolute number of patients positive for SARS-CoV-2 infection (30%); however, those between 40–49 and 70–79 years old had the greatest proportion of positive viral infection, 32.6% and 32.4%, respectively, in comparison with those of 30–39 years old. The 45.4% played a role in the tertiary economic activity. People from the tertiary economic activity presented the highest proportion of SARS-CoV-2 infection compared to primary and secondary (2880; 27.3%), 53.7% did not report employment (Table 1).

Otherwise, in the study population, a total of 10,220 (52.4%) reported an underlying disease, while 9293 (40%) did not inform chronic diseases, and 3698 (15.9%) of the data were missing. The 27.8% of the people with an underlying disease had a current infection for SARS-CoV-2; the most prevalent comorbidity identified was obesity, followed by arterial hypertension and diabetes with 4495 (19.4%), 2452 (10.6%), and 1622 (7%), respectively. Patients reporting diabetes exhibited the highest proportion of viral infection with 563 (34.7%) positive results. Although obesity is the most prevalent comorbidity, only 1400 (31.2%) presented SARS-CoV-2 infection. Among patients reporting pulmonary disease and asthma, nearly 31% presented a viral infection, similar to patients with AIDS and arterial hypertension. In contrast, people with hepatic disease and immunosuppression showed the lowest SARS-CoV-2 infection with only 27% of cases (Table 1).

Based on the travel history recording, a large proportion of people visiting the diagnosis center, 6.7%, reported a mobilization to different countries, cities, or counties. From the people who traveled, 95.3% were national mobilization, mainly in the Jalisco state; meanwhile, 4.7% of patients were informed about international travel. North America was the most visited region by people from Western Mexico with 3.6%, followed by Europe with 0.9%. No diagnosis for SARS-CoV-2 infection in international travelers was reported; on the contrary, 1.5% of the national travelers were positive for viral infection (Table 1).

### 3.2. SARS-CoV-2 Distribution during 2020

Despite the first case of COVID-19 in Jalisco state was reported on 14 March 2020, the largest flux of patients in the center was between July–August and October. Nonetheless, the higher positive infection reports were in July, August and December (Figure 2).

### 3.3. Symptoms Frequency Description in the Studied Population

According to the CDC, symptoms like fever, cough, breath shortness, or prior contact with COVID-19 positive patients are risk filters for SARS-CoV-2 infection [6]. Herein, 4051 (17.5%) of the patients evaluated were asymptomatic; but manifested a previous contact with an infected patient (Table 2); while overall, the headache was the most prevalent symptom present in 63.9% (*n* = 14,825) of the studied population, followed by fatigue and dry cough with 56.5% (10,983) and 51.6% (11,986), which have also been the most recurrent in COVID-19 patients. On the other hand, the least frequent symptom was productive cough and difficulty breathing. Among patients positive for viral infection, headaches also rise as the most prevalent symptoms (Figure 3A). In general, women had the largest proportion of symptoms, such as those negative to SARS-CoV-2. Furthermore, as shown in Figure 3B, the same tendency was observed when the data were analyzed only in positive patients. However, when we evaluated the proportion according to the presence of any symptoms by positive/negative RT-PCR results, we observed a higher proportion of positive patients presenting symptoms compared to those negatives (92.3% vs. 78.4%; *p* < 0.001; data not shown).

### 3.4. Association of Demographic Data, Underlying Diseases and SARS-CoV-2 Positive Diagnosis with Symptomatology

People visiting the sentinel center for SARS-CoV-2 diagnosis manifested COVID-19 related symptomatology or contact with a current infected patient. Even though a large proportion of suspected individuals were negative for viral infection, those with a positive result exhibited a higher odd to present symptomatology, with an OR = 3.28 (CI 95% 2.98–3.60; *p* < 0.001). Fever arose as the symptom with the highest odds to be presented by patients with SARS-CoV-2 infection (OR = 1.90; CI 95%: 1.79–1.19; *p* < 0.001), followed by headache and dry cough (OR = 1.88; CI 95%: 1.66–1.88 and OR = 1.75; CI: 1.65–1.85; *p* < 0.001). Herein, sex did not represent a risk for viral infection (OR = 1.01; CI 95% 0.96–1.07; *p* = 0.98) (Figure 4A).

As previously mentioned, 52.4% of the population presented comorbidities that have been related to COVID-19 severity, from which 27.8% were positive for a current SARS-CoV-2 infection (Table 1). In this context, statistical analysis did not show any increased odds for viral infection when compared among negative/positive infection and comorbidity (OR = 1.02, CI 95%: 0.96–1.08; *p* = 0.57). However, individually, diabetes and obesity presented a higher risk for viral infection with OR = 1.27, CI 1.14–1.41 and 1.08, CI 1.01–1.16, respectively (Figure 4B). Notably, the presence of any underlying disease was statistically associated with the symptomatology of people assisting to the center, increasing the risk for manifesting the COVID-19 clinical features with OR = 1.39, CI 95% 1.29–1.50; *p* < 0.001; however, these odds was not increased when the analysis was carried only in patients with viral infection (OR = 1.079, CI 95% 0.79–1.46; *p* = 0.619 (Table 3)).

Additionally, given that previous studies reported that the clinical presentation of COVID-19 in older people tends to be more severe, we evaluated the association of the number of symptoms reported by patients split by age using 60 years old as a cut-off value. Table 3 showed that age highly influences symptomatology. People >60 years old exhibited increased odds for developing typical COVID-19 clinical features (OR = 3.59, CI: 2.10–6.14). Remarkably, in elderly patients, the influence of age on the clinical outcomes (observed by the number of symptoms) increased in advanced age, showing the largest odds in older people.

## 4. Discussion

After one year of pandemic, we described in the present study the clinical and demographic characteristics of 21,211 people assisting the diagnosis center for the molecular diagnosis from the metropolitan area in Jalisco, Mexico. We found that between April–December 2020, around one-third of the assessed people had SARS-CoV-2 infection, mostly adult females, performing a tertiary-economic activity, but not a statistical association for infection risk. The higher infection rate was registered in the Mexican vacation periods in July, August, and December. Furthermore, symptomatology was statistically related to COVID-19 clinical characteristics; likewise, underlying diseases such as diabetes, obesity, and hypertension were also associated with SARS-CoV-2 infection but not with disease severity. Herein, we also found that patients >60 years old infected with SARS-CoV-2 were more prone to disease severity, confirming the increased risk of the disease with age. This study represents the first report to describe SARS-CoV-2 demographic epidemiology during the 2020 COVID-19 epidemic in Western Mexico.

According to the John Hopkins Institute, during 2020, more than 84 million confirmed SARS-CoV-2 cumulative cases in the world were reported [3]. This institute estimates that in China, where SARS-CoV-2 was first detected, throughout 2020 were reported around 96 thousand confirmed cases of SARS-CoV-2 infection, representing 0.007% of the overall population [3]. Meanwhile, COVID-19 prevalence in México is 1.1%, with over 1.4 million infection cases [13], a significantly higher prevalence than China. Notably, in our study, we found an incidence of 29.8%, which represents 0.14% of the Jalisco state population. Several preventive methods have been implemented since the detection of the first cases in Jalisco, impacting the viral distribution. The first general lockdown was implemented on 24 March, preventing a rapid viral distribution in the next month. However, as shown in our study, an increase in the SARS-CoV-2 infection cases was observed in July (see Figure 2), which is in accordance with the augment of people mobility according to the Google Region Mobility Report [14]. Thereafter, the subsequent lockdown in October did not show changes either in the mobility population [14] or in the number of positively diagnosed infections. During the current year was implemented an exhaustive vaccination program, having an impact on the decrease of the lethality rate [13]. Further analysis should be performed in order to show the behavior of the COVID-19 during 2021 under the schemes of the new SARS-CoV-2 variants and the confrontation with the vaccination programs.

Unlike the country’s general statistics, in our studied population, we found a higher proportion of women with a positive SARS-CoV2 result (53.8%) compared to men (46.2%). This subtle discrepancy was unexpected; however, this is in line with the global statistics, where 51% of the confirmed cases are female and 49% men [15]. It is important to mention that this trend varies from country to country. Some of the identified nations that follow the bias we observed in our population are Belgium, Portugal, France, Switzerland, and South Korea. On the other hand, countries such as Singapore, Bahrain, and Qatar show opposite shifts where 88.93%, 88.42%, and 84.85% of the confirmed cases are males, countries that have in common a bridging gender gap that might explain this differences in the highest men proportion [15]. Overall, most countries share a near 50/50 proportion in sex-disaggregated confirmed cases. The observed variations are not solely explained by biology; other factors specific to the sex, such as behavior and society, play a pivotal role in the obtained epidemiological information. It is important to mention that even though the proportion of confirmed cases disaggregated by sex depend on other factors rather than biology, sex does have a strong influence on the severity of the disease [16].

Some of the social factors that have been reported to affect the proportion of women with a positive result are their participation in certain economic sectors that involved more exposure to the virus and a significant reduction in social and economic security. In detail, women represent the majority of the healthcare sector and care-related activities, which means they are tested more often [17]. Moreover, no statistical differences in SARS-CoV-2 diagnosis were found across the year in the context of economic activity. The highest proportion of the population belongs to tertiary activity. Furthermore, most women from Jalisco state work in this economic activity sector [18]. Additionally, according to global and regional data, the economic impact was unequal; female job loss rates due to COVID-19 are about 1.8 times higher than male job loss rates globally, at 5.7 percent versus 3.1 percent, respectively [19]. Notably, the tertiary sector in Jalisco state is mostly made up of young people between the ages of 25 to 45 with an average age of 28 years [12]. In the present study, we found that the mean age of the population working in the tertiary economic sector was 37 ± 14 years, representing an older population compared to the data reported by the INEGI. However, this might be overrepresented because patients <20 years represent a low proportion of the population assisting for viral diagnosis.

The mean age found in the study population was in line with previous studies in México at the beginning of the pandemic and those recent reports, showing a mean age ranging from 37 to 46 years [20,21,22]. This increased susceptibility in older patients to viral infection has been well documented. For example, a multinational study from China, Italy, Japan, Singapore, Canada, and South Korea estimated that susceptibility for infection in patients <20 years was around half compared to those >20 years old. This difference was also associated with the manifestation of the clinical symptoms [23], which might respond to the higher rate of comorbidities in older patients [24] and the higher susceptibility for clinical manifestation [23].

On the other hand, even though most patients in this study manifested respiratory symptoms, we found a statistical association of those clinical outcomes with SARS-CoV-2 infection. According to the CDC, fever, cough, shortness of breath, and fatigue are the most common symptoms related to COVID-19 [6]; reports from WHO [25] corroborate this information. A systematic review that includes 24,410 COVID-19 confirmed adult patients found that 78% of people presented fever, 57% cough, and 31% fatigue [26], similarly to those reported by da Rosa et al. [27]. Meanwhile, a national epidemiology study in the Mexican population showed that headache was the most common symptom among patients positive to SARS-CoV-2 [21]. Headache, along with dry cough and fever, were also found in a higher proportion in the confirmed cases of our database. It is important to point out that at the time of the molecular diagnosis, the people analyzed here were ambulatory and did not require hospitalization, diminishing the symptoms spectrum and disease severity.

Clinical manifestation of COVID-19 is favorably associated with host characteristics. As previously mentioned, age plays an important role in this context, in which people over 50-year-old have an increased risk for fatal complications for COVID-19 [28]. Sex is a demographic characteristic that mainly influences clinical outcomes. A meta-analysis from more than 3 million global cases found that, in terms of proportion, there were no differences in sex and COVID-19; nonetheless, the author found that males have three times the odds of requiring intensive treatment unit (OR = 2.84; CI = 2.06–3.92) and death (OR = 1.39; CI = 1.31–1.47) [29]. These differences are multifactorial due to disparities in the innate and adaptive immune response [30], production of interferons [31,32], number of CD4+ and CD8+ T cells [32,33], and hormone levels [34]. However, we did not find differences among symptomatology and sex, probably due to the demographical characteristics mentioned above, although in Mexican studies, a male predisposition for disease severity was reported [20,21,22].

Finally, we found that diabetes, obesity, and arterial hypertension were associated with SARS-CoV-2 infection. According to the Federación Mexicana de Diabetes, A.C., heart diseases and diabetes are the first and second place in mortality in the country [35]. Diabetes prevalence in the country is present in 10.7%, while 72.7% are overweight or obese [8], making them more likely to be infected with SARS-CoV-2 and have the worst clinical outcomes. At the pandemic beginning in México, a national analysis found hypertension in 43.5%, diabetes in 39.3%, and obesity in 30.4% of the deceased patients by COVID-19 [20]. Similar results were reported by Fernandez-Rojas et al. [21]. Among the confirmed cases here evaluated, obesity, arterial hypertension, and diabetes were the most prevalent, which is in line with the previously reported results [20]. However, no statistical association was found between comorbidities and disease severity. Accordingly, a recent systematic review and meta-analysis including 345 studies did not find statistical associations among underlying diseases and COVID-19 severity, even when a high prevalence of underlying diseases was found [36]. In addition, survival analysis in México found a hazard ratio = 1.9 in the context of chronic kidney diseases, but not for diabetes or obesity [22]. Otherwise, a meta-analysis at the beginning of the pandemic showed a slight increase in disease severity. Altogether, this information raises the relevance of data collection of comorbidities prevalence on patients with COVID-19 still crucial due to disease influence severity reported previously [37].

## 5. Conclusions

In summary, this study represents the first concise description of the incidence of SARS-CoV-2 in Western Mexico during 2020 that gives information about the viral epidemiological behavior from patient samples assisting sentinel center for molecular diagnosis. We found a high incidence of SARS-CoV-2 and, although females were the prevalent population in our study, no infection or disease severity predisposition was found. Diabetes, obesity, and arterial hypertension were comorbidities associated with a higher risk of infection, but not clinical severity. Otherwise, elderly patients (>60 years old) showed the largest predisposition for molecular positivity to SARS-CoV-2 and clinical severity of COVID-19, which might be augmented according to increasing age. The information presented here can potentially be used as an epidemiological and demographic viral and host characteristic reference for future SARS-CoV-2 outbreaks or another epidemic etiological agent that helps health officials identify patients on risk and elaborate planning efforts of the disease management.

## Figures and Tables

**Figure 1 ijerph-18-10644-f001:**
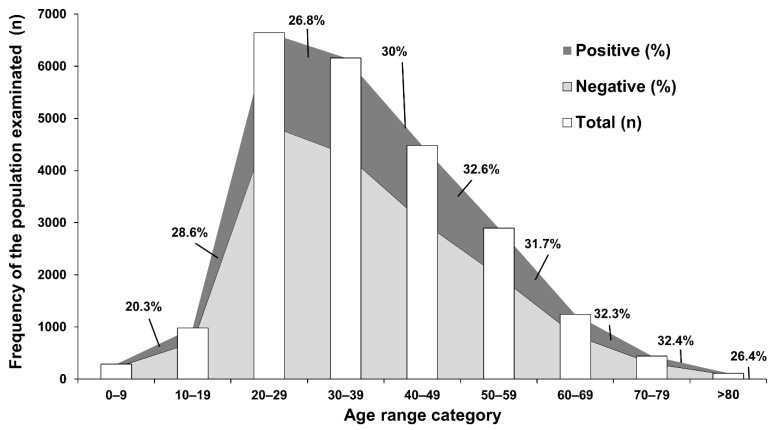
The SARS-CoV-2 epidemiological description along April–December 2020. The largest proportion of patients was represented by group among 20–29 years old, but those in the 40–49, 70–79, and 40–69 groups showed the greatest proportion of positive patients to SARS-CoV-2.

**Figure 2 ijerph-18-10644-f002:**
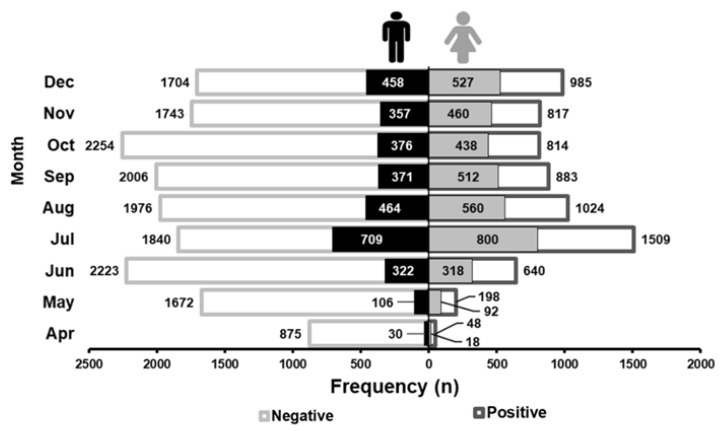
Total of the studied population segregated by sex and molecular SARS-CoV-2 result. The frequency in confirmed cases was increased during Mexican vacation periods (July and December). Number of men and women confirmed cases are shown in black and light gray, respectively. Total frequency of negative and positive cases is shown on the left and right edges, respectively.

**Figure 3 ijerph-18-10644-f003:**
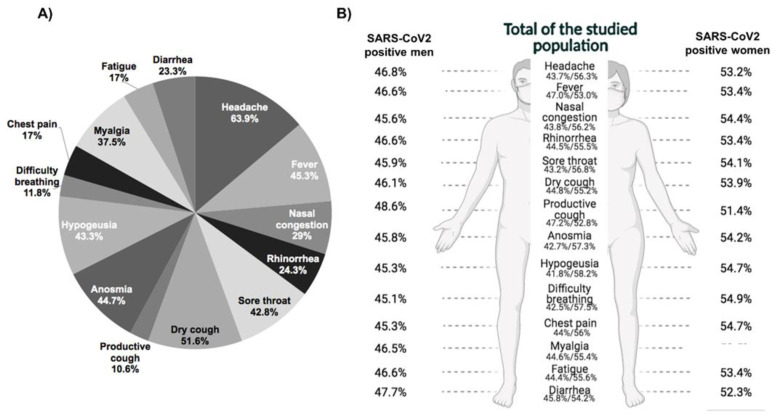
The most common symptoms reported related to COVID-19. (**A**) Symptoms reported by all of the studied population, each slice represents the proportion of the reported. The most common symptoms were: headache, dry cough, and fever. (**B**) Information segregated by sex, values in the middle represent the total of the studied population separated by sex. The proportions shown in both edges constitute only the confirmed cases segregated by sex. In the confirmed cases, more women reported having symptoms than men.

**Figure 4 ijerph-18-10644-f004:**
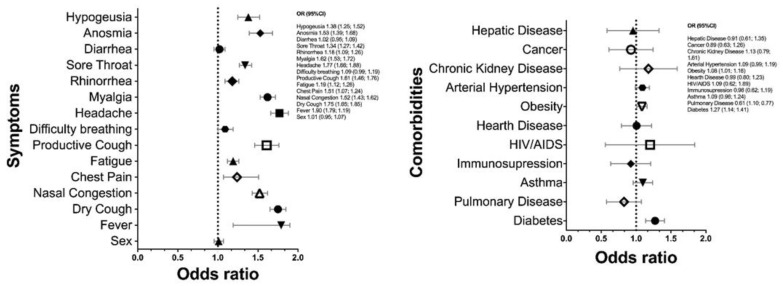
Symptoms and comorbidities are highly associated with SARS-CoV-2 infection. (**A**-**the left**) Fever, headache, and dry cough were the most characteristics symptoms among patients with SARS-CoV-2 infection; sex did not exhibit an association with the presence of the virus. Meanwhile, (**B**-**the right**) the underlying diseases that demonstrated a positive relation with SARS-CoV-2 were diabetes, obesity, and hypertension.

**Table 1 ijerph-18-10644-t001:** Demographic, clinical, underlying diseases, and travel history from study population among April–December 2020.

	Total *n* (%)	SARS-CoV-2 Negative *n* (%)	SARS-CoV-2 Positive *n* (%)	*p*-Value
*n* (%)	23,211	16,293 (70.2%)	6918 (29.8%)	<0.001
Age (mean ± SD)	37.4 ± 14.2	37.0 ± 14.2	38.4 ± 13.9	NS
Sex				NS
Male	10,761 (46.4%)	7568 (70.3%)	3193 (29.7%)	<0.001
Female	12,450 (53.6%)	8725 (70.1%)	3725 (29.9%)	<0.001
Economic Activity				<0.001
No data	12,515 (53.7%)	8520 (68.1%)	3995 (31.9%)	<0.001
Primary	17 (0.1%)	13 (76.5%)	4 (23.5%)	0.029
Secondary	149 (0.8%)	110 (73.8%)	39 (26.2%)	<0.001
Tertiary	10,530 (45.4%)	7650 (72.7%)	2880 (27.4%)	<0.001
Underlying Disease	10,220 (52.4%)	7376 (72.2%)	2844 (27.8%)	<0.001
Diabetes	1622 (7%)	1059 (65.3%)	563 (34.7%)	<0.001
Pulmonary disease	135 (0.6%)	92 (68.2%)	43 (31.9%)	<0.001
Asthma	1209 (5.2%)	819 (67.7%)	390 (32.3%)	<0.001
Immunosuppression	183 (0.8%)	132 (72.1%)	51 (27.9%)	<0.001
HIV/AIDS	57 (0.2%)	39 (68.42%)	18 (31.6%)	<0.001
Hearth disease	417 (1.8%)	296 (71%)	121 (29.0%)	<0.001
Obesity	4495 (19.4%)	3095 (68.9%)	1400 (31.2%)	<0.001
Arterial hypertension	2452 (10.6%)	1673 (68.2%)	779 (31.8%)	<0.001
Chronic kidney disease	136 (0.6%)	92 (67.7%)	44 (32.4%)	<0.001
Cancer	155 (0.8%)	110 (71%)	45 (29%)	<0.001
Hepatic disease	122 (0.5%)	89 (73%)	33 (27%)	<0.001
Travel history	1567 (6.7%)			
National	1493 (95.3%)	1471 (98.5%)	22 (1.5%)	<0.001
International	74 (4.7%)	74 (100.00%)	-	
North America	56 (3.6%)	56 (100.00%)	-	
South America	1 (0.1%)	1 (100.00%)	-	
Europe	14 (0.9%)	14 (100.00%)	-	
Asia	2 (0.1%)	2 (100.00%)	-	
Oceania	1 (0.1%)	1 (100.00%)	-	

**Table 2 ijerph-18-10644-t002:** Reported symptoms from people assisting to monitoring diagnosis center for molecular SARS-CoV-2 testing; all individual symptoms and lack of symptoms (asymptomatic) are included.

	Total (*n* = 23,211) *n* (%)	SARS-CoV-2 Negative *n* (%)	SARS-CoV-2 Positive *n* (%)	*p*-Value
Asymptomatic	4051 (17.5%)	3515 (86.8%)	536 (13.2%)	<0.001
Headache	14,825 (63.9%)	9811 (66.2%)	5014 (33.8%)	<0.001
Fever	10,513 (45.3%)	6630 (63.1%)	3883 (36.9%)	<0.001
Nasal congestion	6722 (29%)	4301 (64%)	2421 (36%)	<0.001
Rhinorrhea	4729 (24.3%)	3118 (65.9%)	1611 (34.1%)	<0.001
Sore throat	9936 (42.8%)	6623 (66.7%)	3313 (33.3%)	<0.001
Dry cough	11,986 (51.6%)	7751 (64.7%)	4235 (35.3%)	<0.001
Productive cough	2061 (10.6%)	1214 (58.9%)	847 (41.1%)	<0.001
Anosmia	3467 (44.7%)	2131 (61.5%)	1336 (38.5%)	<0.001
Dysgeusia	3361 (43.3%)	2104 (62.6%)	1257 (37.4%)	<0.001
Difficulty breathing	2731 (11.8%)	1876 (68.7%)	855 (31.3%)	<0.001
Chest pain	3941 (17%)	2677 (67.9%)	1264 (32.1%)	<0.001
Muscle pain	8703 (37.5%)	5555 (63.8%)	3148 (36.2%)	<0.001
Fatigue	10,983 (56.5%)	7361 (67.1%)	3622 (33%)	<0.001
Diarrhea	5402 (23.3%)	3795 (70.3%)	1607 (29.8%)	<0.001

**Table 3 ijerph-18-10644-t003:** Risk factors associated with the COVID-19 clinical presentation.

	Asymptomatic *n* (%)	1–3 Symptoms *n* (%)	4–5 Symptoms *n* (%)	>6 Symptoms *n* (%)	*p*-Value	OR (IC 95%)
Age (>60)	14 (2.6%)	144 (10.1%)	242 (9.5%)	174 (7.3%)	<0.001	3.59 (2.10–6.14)
Age range					<0.001	1.10 (1.04–1.18)
0–9	73 (1.8%)	90 (1.8%)	76 (1.0%)	47 (0.7%)		
10–19	137 (3.4%)	216 (4.4%)	343 (4.7%)	283 (4.2%)	<0.001	4.92 (2.27–10.70)
20–29	1164 (28.7%)	1378 (27.7%)	2099 (28.5%)	1995 (29.3%)	<0.001	4.48 (2.39–8.41)
30–39	1331 (32.7%)	1222 (24.6%)	1853 (25.1%)	1740 (25.5%)	<0.001	2.96 (1.59–5.52)
40–49	818 (20.2%)	924 (18.6%)	1384 (18.8%)	1350 (19.8%)	<0.001	3.34 (1.78–6.26)
50–59	375 (9.3%)	633 (12.7%)	972 (13.2%)	914 (13.4%)	<0.001	4.04 (2.11–7.75)
60–69	113 (2.8%)	346 (7%)	437 (5.9%)	343 (5%)	<0.001	11.25 (4.81–26.30)
70–79	30 (0.7%)	121 (2.4%)	173 (2.4%)	121 (1.8%)	<0.001	22.75 (4.98–103.98)
>80	10 (0.2%)	37 (0.7%)	37 (0.5%)	26 (0.4%)	<0.001	8.91 (1.11–71.55)
Sex	536 (7.75%)	–	–	–	0.194	0.89 (0.74–1.06)
No. of comorbidities					0.619	1.08 (0.079–1.46)
0	103 (19.2%)	748 (52.5%)	1363 (53.2%)	1310 (54.7%)		
1–3	54 (10.1%)	519 (36.4%)	1025 (40%)	958 (40%)	0.12	1.396 (0.85–1.95)
>4	379 (70.7%)	158 (11.1%)	173 (6.8%)	128 (5.3%)	0.16	0.36 (0.029–1.53)

## Data Availability

The data presented in this study are available on request from the corresponding author. The data are not publicly available due to data storage policy.

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
