# Peer review of "A Comprehensive Descriptive Epidemiological and Clinical Analysis of SARS-CoV-2 in West-Mexico during COVID-19 Pandemic 2020"

_ijerph, 2021, doi:10.3390/ijerph182010644_

Round 1

Reviewer 1 Report

This manuscript of Oliver Viera-Segura et al. describes the epidemiological situation of SARS-CoV-2 pandemia in Mexico. This data are important, as COVID-19 infection represents an important threat to the global health system.

Nevertheless, there are several data that should be included:

  1. There is no data concerning the genetic variability of the virus. 
  2. There is no data about how many patients were treated ambulatory, how many were hospitalized (including how many required intensive care unit admission) 
  3. There is no data concerning the mortality (cause of death, % of death....)
  4. The information about treatment programmes should be added.
  5. Analysis of the risk factor for severe COVID-19 infection
  6. What type of preventive methods are introduced by the Mexican government? Has it changed the pandemic situation?

Author Response

 RE: Manuscript ID: ijerph-1398288

Enclosed is a complete copy of the revised version of the manuscript, entitled A comprehensive descriptive epidemiological and clinical analysis of SARS-CoV-2 in West-Mexico during COVID-19 pandemic 2020” by Drs. Viera-Segura et al., which is being re-submitted for publication in your prestigious journal.

We appreciated the thorough and critical analyses by the reviewers. The manuscript has been fully revised in accordance with the Reviewer´s and Editor´s comments. All issues raised by reviewers were addressed as recommended and are marked in the revised version.

We hope that this manuscript is now acceptable for publication in International Journal of Environmental Research and Public Health.

Sincerely.

Reviewer´(s) comments to the author:

Reviewer #1:

This manuscript of Oliver Viera-Segura et al. describes the epidemiological situation of SARS-CoV-2 pandemia in Mexico. This data are important, as COVID-19 infection represents an important threat to the global health system.

Comments for transmission to the author.

Nevertheless, there are several data that should be included:

Q1.- There is no data concerning the genetic variability of the virus.

Q2.- There is no data about how many patients were treated ambulatory, how many were hospitalized (including how many required intensive care unit admission)

Q3.- There is no data concerning the mortality (cause of death, % of death....

Q4.- The information about treatment programmes should be added.

Q5.- Analysis of the risk factor for severe COVID-19 infection.

A1= We highly appreciate the observation of the reviewer in order to improve the information presented here. It is important to point out that LaDEER is an ambulatory diagnosis center designed for the Universidad de Guadalajara for social support in the epidemiological COVID-19 monitoring. As an ambulatory laboratory, we did not have access to the complete clinical record of the patients assisting the center for SARS-CoV-2 molecular diagnosis. For this reason, the information concerning the medical intake, hospitalization proportion, ICU admission, disease severity, or death proportion can not be obtained. As stipulated in the methods section, lines 97-105, all the information included herein was obtained transversely prior to sample collection through a telephonic survey in collaboration with health institutions, and no clinical follow-up was performed. Some modifications in this regard were included in the M&M section.

Furthermore, as the reviewer appointment, the analysis about the genetic variability of the virus is very interesting; however, it scapes from our aims since the nature of the LaDEER function is the SARS-CoV-2 diagnosis, no further scrutiny in this regard could be performed.

Q6.- What type of preventive methods are introduced by the Mexican government? Has it changed the pandemic situation?

A2=This question is very interesting. The Mexican government has implemented several preventive methods during the COVID-19 pandemic; in this regard, we add a segment on this subject in the discussion section (lines 351-362).

The authors appreciate the throughout and critical analysis by the reviewers.

Reviewer 2 Report

Review „A comprehensive descriptive epidemiological and clinical analysis of SARS-CoV-2 in West-Mexico during COVID-19 pandemic 2020“

Thank you for the chance to review the article cited above. Oliver Viera-Segura and his colleagues report a high SARS-CoV-2 prevalence in Western Mexico. According to experience, comorbidities were frequent in infected people; nevertheless, no association with disease outcomes was observed, in contrast with the highest disease severity risk found in older patients, which in fact is not astonishing. As expected, the authors conclude, that continuous monitoring should be carried since comorbidities have been reported as aggravating factors. This study was intended to help health officials for the elaboration of planning efforts of the disease management and others in the future.

Language style and grammar are acceptable; consider revision on the one or another spot.

Basically, their findings are in line with all recent published results and do not necessarily offer overwhelming news for the scientific community.

Below, you will find some minor points of criticism:

Introduction

General: Consider shortening as etiology and pathogenesis have been reported extensively elsewhere. Focus on Mexico.

Line 60:

“After an incubation period of 1-14 days, COVID-19 shows a mild course in 80 % of 60 observed patients and a severe course in 20%, with a 0.3-5.8 % lethality rate.” Consider adding overall lethality rate, since lethality may be much higher on ICU.

Line 65:

“As of August, there were 3,225,073 confirmed cases and 253,155 deaths in México, while 127,023 cases and 14,202 deaths were reported in Jalisco state.”

The lethality rate (i. e. 7,9 – 11,2%) appears to be higher than the previously reported average – please explain.

Material and methods:

Is there an ethics approval? Registration in an international trial register? Informed consent?  

2.3 Statistical analysis

Which p was determined to be of statistical significance?

Results:

Line 139: “It is important to point out that even when patients with age ranging from 30-39 showed the largest absolute number of patients positive for SARS-CoV-2 infection (29.96%), those between 40-49 and 70-79 years old had the greatest proportion of positive 141 viral infection, 32.58% and 32.43%, respectively.”

Only one decimal place - please round. This sentence is judgmental – consider reviews.

Line 142: “The economic activity was classified into sectors according to the Instituto Nacional de Estadística y Geografía (12) into primary as the exploitation of natural resources as forestry, agriculture, mining, and quarrying; secondary as manufacturing, processing, and construction (infrastructure) industries; and tertiary as goods distribution, information transactions, transactions with assets, services whose primary input is the knowledge and experience of personnel, recreation-related services, and other (12).”

In my believe, this is part of the M&M section (or discussion) – consider revision.

Line 193: “It is important to point out that at the time of the molecular diagnosis, the people analyzed here are ambulatory and did not require hospitalization, diminishing the symptoms spectrum and disease severity.”

In my believe, this is part of the M&M section (or discussion) – consider revision.

Line 206: COVI-19. A “D” is missing.

Quality of figure 4 is insufficient – a higher resolution is needed.

Discussion

Line 348: Abbrevations (HZ) need to be explained.

Author Response

RE: Manuscript ID: ijerph-1398288

Enclosed is a complete copy of the revised version of the manuscript, entitled A comprehensive descriptive epidemiological and clinical analysis of SARS-CoV-2 in West-Mexico during COVID-19 pandemic 2020” by Drs. Viera-Segura et al., which is being re-submitted for publication in your prestigious journal.

We appreciated the thorough and critical analyses by the reviewers. The manuscript has been fully revised in accordance with the Reviewer´s and Editor´s comments. All issues raised by reviewers were addressed as recommended and are marked in the revised version.

We hope that this manuscript is now acceptable for publication in International Journal of Environmental Research and Public Health.

Sincerely.

 Reviewer´(s) comments to the author:

Reviewer #2:

Review “A comprehensive descriptive epidemiological and clinical analysis of SARS-CoV-2 in West-Mexico during COVID-19 pandemic 2020“

Thank you for the chance to review the article cited above. Oliver Viera-Segura and his colleagues report a high SARS-CoV-2 prevalence in Western Mexico. According to experience, comorbidities were frequent in infected people; nevertheless, no association with disease outcomes was observed, in contrast with the highest disease severity risk found in older patients, which in fact is not astonishing. As expected, the authors conclude, that continuous monitoring should be carried since comorbidities have been reported as aggravating factors. This study was intended to help health officials for the elaboration of planning efforts of the disease management and others in the future.

Language style and grammar are acceptable; consider revision on the one or another spot.

Basically, their findings are in line with all recent published results and do not necessarily offer overwhelming news for the scientific community.

Below, you will find some minor points of criticism:

Comments for transmission to the author.

Introduction

Q1.-General: Consider shortening as etiology and pathogenesis have been reported extensively elsewhere. Focus on Mexico.

A1.- As suggested by the reviewer, the introduction method was adapted to focus the information on Mexico.

Line 60:

“After an incubation period of 1-14 days, COVID-19 shows a mild course in 80 % of observed patients and a severe course in 20%.

Q2.- Consider adding overall lethality rate, since lethality may be much higher on ICU.

A2.- We recognize this text was not clear enough to be interpreted; therefore, we changed the sentence for a better understanding and can be found in  lines 59-69.

Q3.- Line 65: “As of August, there were 3,225,073 confirmed cases and 253,155 deaths in México, while 127,023 cases and 14,202 deaths were reported in Jalisco state.”

Q: The lethality rate (i. e. 7,9 – 11,2%) appears to be higher than the previously reported average – please explain.

A3.- We really appreciate this comment since the information could be misunderstood. The text was restructured for better understanding, lines 64-69. 

Material and methods:

Q4: Is there an ethics approval? Registration in an international trial register? Informed consent?  

A4.- Thank you for this observation. We recognize that ethics approval is crucial for high-quality researches, especially on this topic; therefore, we included the requested information in lines 466-472.

2.3 Statistical analysis

Q5: Which p was determined to be of statistical significance?

A5.- We acknowledge that the p-value to determine statistical significance was not written; therefore, we really appreciate this statement. In this regard, the p-value was included in line 135.

Results:

Line 139: “It is important to point out that even when patients with age ranging from 30-39 showed the largest absolute number of patients positive for SARS-CoV-2 infection (29.96%), those between 40-49 and 70-79 years old had the greatest proportion of positive 141 viral infection, 32.58% and 32.43%, respectively.”

Q6: Only one decimal place - please round. This sentence is judgmental – consider reviews.

A6.-The data was changed to one decimal place, according to the reviewer appointment. As to the sentence, we restructured as you suggested, so it only presents the data found, and no judgment is presented in the section; you may find this change in lines 144-148.

Line 142: “The economic activity was classified into sectors according to the Instituto Nacional de Estadística y Geografía (12) into primary as the exploitation of natural resources as forestry, agriculture, mining, and quarrying; secondary as manufacturing, processing, and construction (infrastructure) industries; and tertiary as goods distribution, information transactions, transactions with assets, services whose primary input is the knowledge and experience of personnel, recreation-related services, and other (12).”

Q7: In my believe, this is part of the M&M section (or discussion) – consider revision.

A7.- Thank you for this observation. We agree with the reviewer, and according to the observation, we replaced the economic activity classification to M&M section in lines 107-112.

Line 193: “It is important to point out that at the time of the molecular diagnosis, the people analyzed here are ambulatory and did not require hospitalization, diminishing the symptoms spectrum and disease severity.”

Q8: In my believe, this is part of the M&M section (or discussion) – consider revision.

A8.- In order to provide better fluidity and coherence to the Results and Discussion sections, we replaced the sentence to the discussion section and can be reviewed in lines 411-414.

Line 206: COVI-19.

Q9: “D” is missing.

A9.- Thank you for pointing this out; the letter D has been written down to complete the word in line 271.

Q10: Quality of figure 4 is insufficient – a higher resolution is needed.

A10.- Major change in the image resolution (600 dpi) of Figure 4 was carried out.

Discussion

Q11: Line 348: Abbrevations (HZ) need to be explained.

A11.- We appreciate this observation. The abbreviation of “HZ” was misspelled in the text and it has been corrected to “hazard ratio” in line 442.

The authors appreciate the throughout and critical analysis by the reviewers.

Round 2

Reviewer 1 Report

In my opinion,  manuscript, entitled A comprehensive descriptive epidemiological and clinical analysis of SARS-CoV-2 in West-Mexico during COVID-19 pandemic 2020” by Drs. Viera-Segura et al., is now acceptable for publication in International Journal of Environmental Research and Public Health.

Now it is clearly explanied that we are speaking about abmulatory patients, and data such as: molecular variability, severity of the patients are not accesible.

Moreover, curently authors included information concerning preventing methodes introduced by health officials in Mexico.